# Exceptional catalytic activity of oxygen evolution reaction via two-dimensional graphene multilayer confined metal-organic frameworks

Siliu Lyu[1,2,8], Chenxi Guo [3,8], Jianing Wang[2], Zhongjian Li[1,4], Bin Yang[1,4], Lecheng Lei [1,4], Liping Wang[2,5], Jianping Xiao [3,5,6] ✉, Tao Zhang [2,5,7] ✉ & Yang Hou [1,4] ✉

Oxygen evolution reaction (OER) plays a key role in many renewable energy technologies such as water splitting and metal-air batteries. Metal-organic frameworks (MOFs) are appealing to design efficient OER electrocatalysts, however, their intrinsic poor conductivity strongly hinders the activity. Here, we show a strategy to boost the OER activity of poor-conductive MOFs by confining them between graphene multilayers. The resultant NiFe-MOF//G gives a record-low overpotential of 106 mV to reach 10 mA cm$^{-2}$ and retains the activity over 150 h, which is in significant contrast to 399 mV of the pristine NiFe-MOF. We use X-ray absorption spectroscopy (XAS) and computations to demonstrate that the nanoconfinement from graphene multilayers not only forms highly reactive NiO$_6$-FeO$_5$ distorted octahedral species in MOF structure but also lowers limiting potential for water oxidation reaction. We also demonstrate that the strategy is applicable to other MOFs of different structures to largely enhance their electrocatalytic activities.

Electrochemical conversion of water in virtue of electrocatalysts to produce eco-friendly and sustainable hydrogen energy source has been widely studied for decades[1]. As the bottleneck of water splitting, oxygen evolution reaction (OER) at the anode requires a relatively large thermodynamic potential (over 1.23 V vs. reversible hydrogen electrode, RHE) to overcome the sluggish kinetics due to its four 'electron-proton' transfer processes[2–5]. Noble metal-based catalysts such as IrO$_2$ and RuO$_2$ have been proved to be superior catalysts for OER[6,7]. However, the scarcity, high cost, and low stability preclude their widespread applications[8]. In this regard, it is highly desirable to develop

low-cost noble metal-free materials as alternative OER electrocatalysts. Transition metal (Ni, Fe, Co, etc.) based materials are known to exhibit high electrocatalytic activity towards OER, however, the easy change of the number and location of active sites for transition-metal oxides, hydroxides, oxyhydroxides and their derivatives is hard to alleviate[9–12].

Recently, metal-organic frameworks (MOFs) emerged as promising candidates for OER electrocatalysts owing to their large surface area, tunable porosity, as well as diverse compositions and metal centers[11,13]. Despite numerous MOF materials have been developed to catalyze OER, some essential issues remain to be solved. For example, most of the

[1]Key Laboratory of Biomass Chemical Engineering of Ministry of Education, College of Chemical and Biological Engineering, Zhejiang University, 38 Zheda Road, Hangzhou 310027, China. [2]Ningbo Institute of Materials Technology and Engineering, Chinese Academy of Sciences, 1219 Zhongguan West Road, Ningbo 315201, China. [3]State Key Laboratory of Catalysis, Dalian Institute of Chemical Physics, Chinese Academy of Sciences, Dalian National Laboratory for Clean Energy, 457 Zhongshan Road, Dalian 116023, China. [4]Institute of Zhejiang University-Quzhou, 78 Jiuhua Boulevard North, Quzhou 324000, China. [5]University of Chinese Academy of Sciences, Beijing 100049, China. [6]Dalian National Laboratory for Clean Energy, Dalian 116023, China. [7]Donghai Laboratory, Room 215, Administration Building, No.1 Zheda Road, Zhoushan 316021, China. [8]These authors contributed equally: Siliu Lyu, Chenxi Guo. ✉e-mail: xiao@dicp.ac.cn; tzhang@nimte.ac.cn; yhou@zju.edu.cn

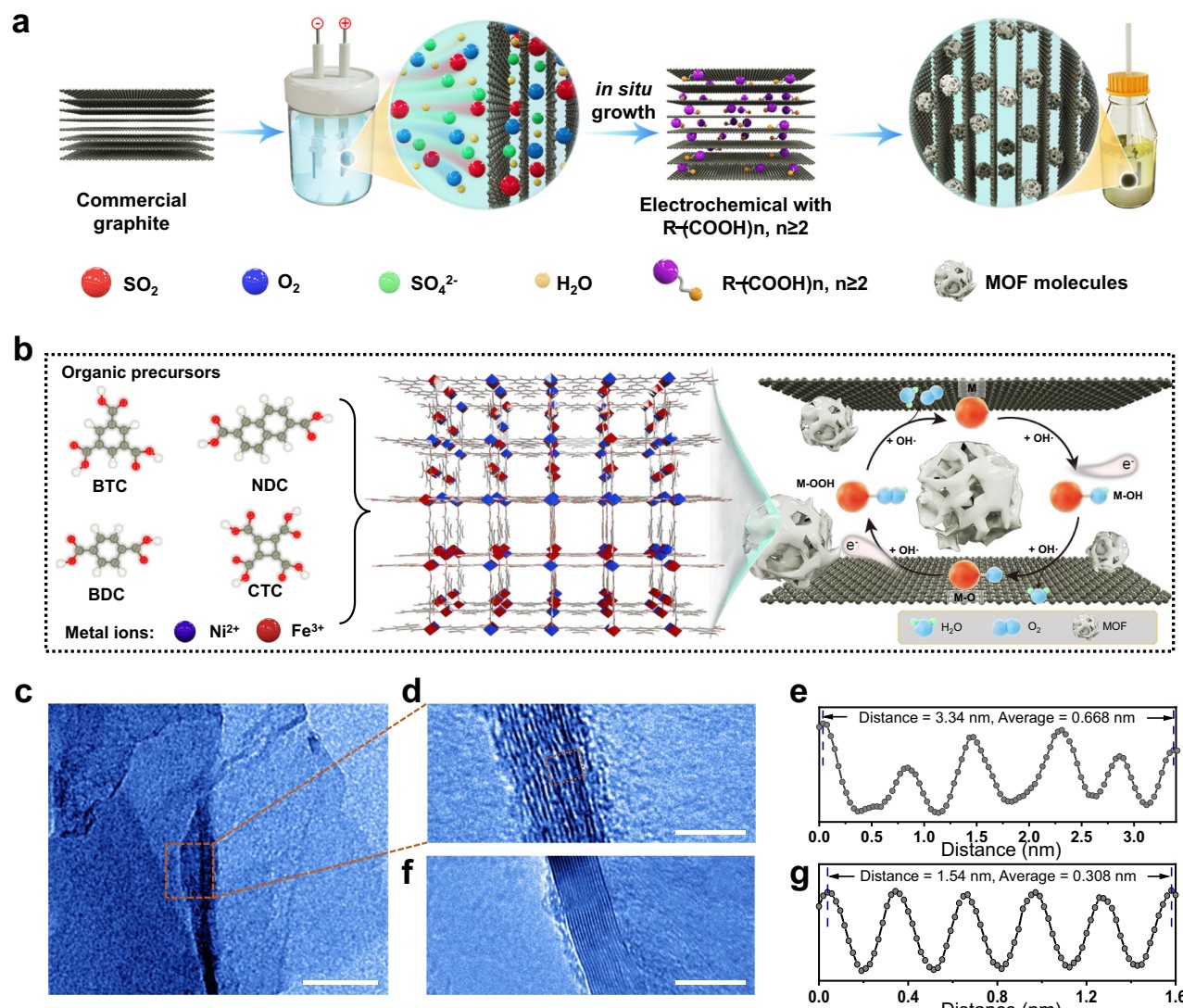

**Fig. 1 | Synthesis and characterizations.** Schematic illustration of **a** the electrochemical synthesis process and **b** the resultant NiFe-MOF//G. **c** TEM image of the cross-section of NiFe-BTC//G. **d** A close-up of (**c**). **f** Cross-sectional view of pristine graphite foil. **e** Calculated interlayer spacings of NiFe-BTC//G and **g** graphite foil. Scale bars, 100 nm (**c**), 5 nm (**d**), 5 nm (**f**).

MOFs are burdened with the intrinsic poor electroconductivity due to the insulating properties of organic ligands and the poor conjugation of metal-organic connection[14–16]. As such, the MOFs based electrodes usually suffer with low mass permeability[17,18]. A variety of methods have been proposed to solve these issues, such as the exfoliation of MOFs into ultrathin layers[19,20], design of complicated linkers to obtain conductive MOFs[21–25], carbonization of MOFs, and so on[26,27]. However, so far the electrocatalytic activities of MOF-based catalysts are still unsatisfied in comparison to state-of-the-art noble metal-based catalysts.

In this work, we show a strategy to strongly enhance the catalytic performance of poorly conductive MOFs by confining them into two-dimensional graphene multilayers. Using this approach, the overpotential of NiFe-BTC surprisingly drops from 399 mV (vs. RHE) to 106 mV (i.e., NiFe-BTC//G) at current density of 10 mA cm$^{-2}$ in 1.0 M KOH, which is superior to state-of-the-art reported MOFs catalysts, and even outcompete noble metal-based catalysts[28–32]. Moreover, the NiFe-BTC//G electrodes are stable and retain the performance for more than 150 h without obvious activity decay. The mechanistic details and active sites of the NiFe-BTC//G are proposed by a combination of X-ray absorption (XAS) experiments and density-functional theory (DFT) calculations.

## Results

### Synthesis and structural characterization

We synthesize the NiFe-BTC//G via a two-electrode electrochemical system (Fig. 1a; Supplementary Movie 1). In brief, a commercial graphite foil was expanded to obtain graphene multilayers in 0.5 M H$_2$SO$_4$ solution for 0.5 h. Subsequently, specific organic salt solution was employed as electrolyte to insert the organic ligands into the dilated graphene/graphite layers[33–36]. The as-treated graphite foil was immersed into metal salt solution (e.g., Ni$^{2+}$ or Fe$^{3+}$) to form MOF intercalations at the interface of graphene multilayers (Fig. 1b). High-resolution transmission electron microscopy (HRTEM) of NiFe-BTC//G from cross-sectional view indicates that the interlayer spacing of graphite foil ($d$ ~ 0.308 nm) increases to ~0.668 nm after the intercalation of MOFs (Fig. 1c–g). Atomic force microscopy (AFM) characterization and energy dispersive X-ray spectroscopy (EDX) demonstrate that NiFe-BTC nanoparticles are uniformly distributed on the surface of multilayer graphene (Supplementary Figs. 1–3). Inductively coupled plasma-mass spectrometry (ICP-MS) determines the precise molar ratios of Ni and Fe species in NiFe-BTC//G with 0.21 wt.% and 6.56 wt.%, respectively (Supplementary Table 1). In addition, three peaks of Raman spectra centered at 485, 557, and 717 cm$^{-1}$ could be attributed

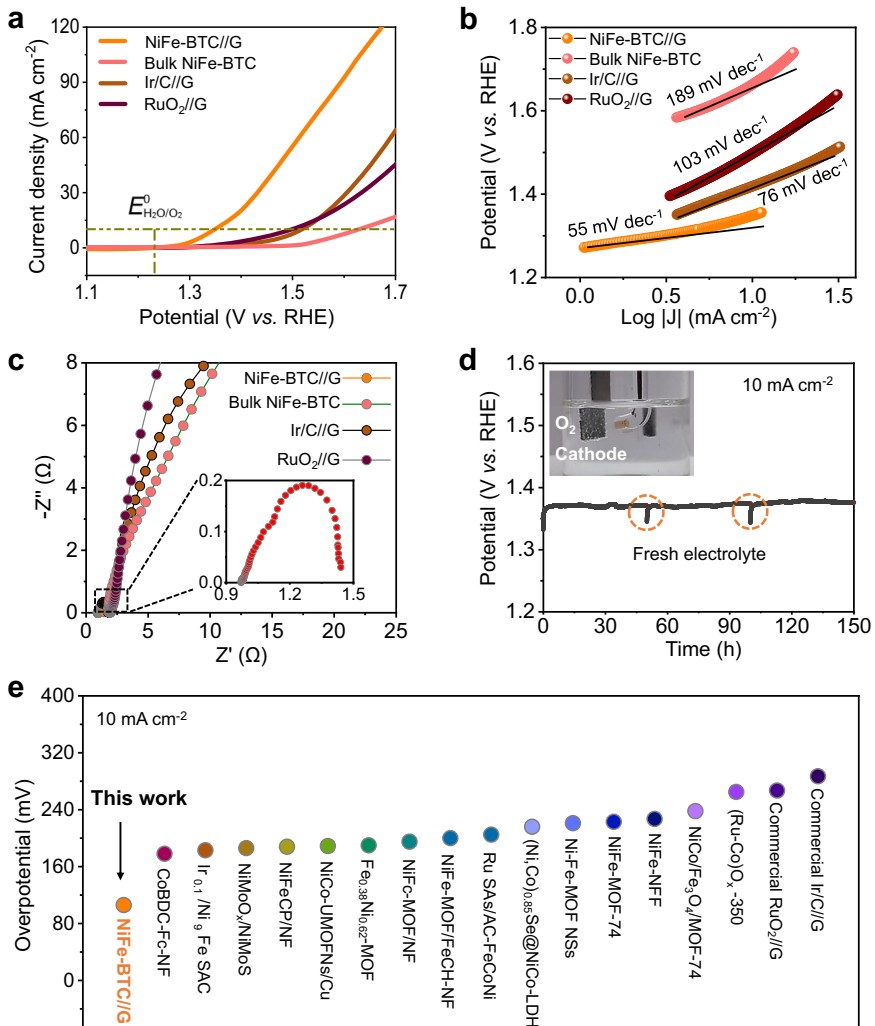

**Fig. 2 | OER electrochemical activity of NiFe-BTC//G. a** LSV plots obtained with NiFe-BTC//G, bulk NiFe-BTC, commercial Ir/C and RuO₂ for OER at 10 mV s⁻¹ in 1.0 M KOH with Ag/AgCl as reference electrode. **b** Tafel plots obtained with NiFe-BTC//G, bulk NiFe-BTC, commercial IrO₂ and RuO₂. **c** EIS Nyquist plots.

**d** Chronopotentiometric testing of NiFe-BTC//G for 150 h at 10 mA cm⁻² in 0.1 M KOH. **e** Comparison of the required voltage at 10 mA cm⁻² for NiFe-BTC//G with other state-of-the-art MOF-based electrocatalysts.

to the characteristic bands of M-O or M-O-M (M represents $Ni^{2+}$ or $Fe^{3+}$) in NiFe-BTC (Supplementary Fig. 4)[37,38]. The peaks at around 1760–1670 cm⁻¹ could be correlated with the coordination of $Ni^{2+}$ or $Fe^{3+}$ ions with BTC ligand[39]. Brunauer-Emmett-Teller (BET) shows that the intercalated NiFe-BTC//G has a typical high specific surface area of 762.7 m² g⁻¹ and total pore volume of 0.15 cm³ g⁻¹ (Supplementary Fig. 5).

**Catalytic performances**

We then use the NiFe-BTC//G directly as electrodes to catalyze the anodic reaction of water splitting in alkaline solution via a typical three-electrode system (Supplementary Figs. 6–12). Both the Ag/AgCl (saturated KCl) with a salt bridge and Hg/HgO are used as the reference electrodes. The noble metal-based Ir/C//G and RuO₂//G catalysts are employed for comparison (Electrode preparation in Supplementary Information). As the linear sweep voltammograms (LSV) shows that the NiFe-BTC//G has a remarkably low overpotential of ~106 mV to achieve benchmark current density of 10 mA cm⁻², which is in significant contrast to bulk NiFe-BTC (399 mV) as well as noble metal-based Ir/C//G (287 mV) and RuO₂//G (267 mV) at identical conditions (Fig. 2a). Such high overpotential of bulk NiFe-BTC is reasonable since its poor electrical conductivity (~5.8 × 10⁻¹² S cm⁻¹) precludes the exposure of catalytic centers as well as efficient mass transports during

catalysis (Supplementary Table 2)[11,40,41]. The confinement environment endows the NiFe-MOF/G-2h electrode stabilized active sites, enhanced electrical conductivity and greatly reduced mass transport lengths. Laying the Fick's law side-by-side with the more comprehensive reports, one can indicate that the concentration of intermediates become gradient in the confinement environments during OER, which propels the intermediates from the center to the boundaries with abundant active sites[22,42,43]. More favorable electrocatalytic kinetic for NiFe-BTC//G can be demonstrated by the smaller Tafel slope of 55 mV dec⁻¹ in comparison to bulk NiFe-BTC powder (189 mV dec⁻¹), and commercial Ir/C//G (76 mV dec⁻¹) and RuO₂//G (103 mV dec⁻¹) (Fig. 2b). The smallest charge transfer resistance ($R_{ct}$ ~ 0.46 Ω) from electrochemical impedance spectroscopy (EIS, Fig. 2c, Supplementary Fig. 13, Supplementary Tables 3, 4) and the largest electrochemical double-layer capacitance ($C_{dl}$~81.6 mF cm⁻²) from the electrochemically active surface area (ECSA, Supplementary Figs. 14, 15) confirm the rapid electron transfer ability as well as highly exposed active sites of NiFe-BTC//G, respectively. Besides of high electrocatalytic activity, NiFe-BTC//G also has good electrocatalytic stability for OER (Fig. 2d, Supplementary Figs. 16–18). As shown in Fig. 2d, NiFe-BTC//G retains its electrocatalytic activity at a current density of 10 mA cm⁻² for 150 h. We note that the current confining strategy is also applicable to other MOFs of different structures to largely

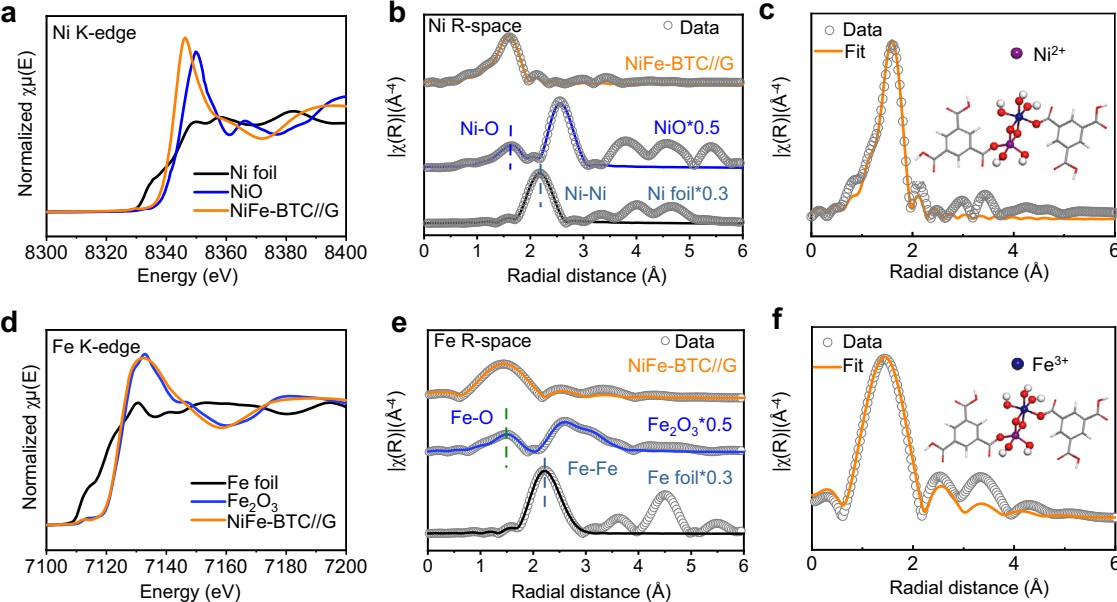

**Fig. 3 | XANES, EXAFS and FT-EXAFS spectra. a** Ni K-edge XANES and **b** extended XANES oscillation functions $k^3 \chi$ (k) with reference samples of Ni foil, NiO. **c** EXAFS fitting curves in R space. Inset: related schematic model of Ni coordination environment in NiFe-BTC//G. **d** Fe K-edge XANES and **e** extended XANES oscillation functions $k^3 \chi$ (k) with reference samples of Fe foil, Fe$_2$O$_3$. **f** EXAFS fitting curves in R space. Inset: related schematic model of Fe coordination environment in NiFe-BTC//G.

improve their electrocatalytic OER activities (Supplementary Fig. 19 and Supplementary Table 5), and the confined BTC structure was confirmed both experimentally and theoretically to perform the highest OER activity in studied MOFs (Supplementary Fig. 19 and Supplementary Fig. 40). We note that the OER activity of the NiFe-BTC//G electrodes outperforms those previously reported MOFs and their derivatives (Fig. 2e, Supplementary Fig. 20; more examples see Supplementary Table 7). The Faradic efficiencies of NiFe-BTC//G at overpotential of 106 mV obtained in 24 h which maintain -100% and the related current densities stay at -10 mA cm$^{-2}$, which confirms that the current corresponds to O$_2$ production with good stability (Supplementary Fig. 21). To verify the low onset potential (1.27 V vs. RHE), the Faradic efficiencies of NiFe-BTC//G at from 1.2 to 1.6 V are investigated, which further confirm the low voltage response of NiFe-BTC//G during OER process (Supplementary Fig. 22). In consideration of different loading amount of NiFe-BTC//G, bulk NiFe-BTC, Ir/C//G, and RuO$_2$//G, the metal mass activity at 1.5 V vs. RHE was adopted to compare their OER performance. The results show that the NiFe-BTC//G could reach the highest mass activity of 111.643 A g$^{-1}_{Ni,Fe}$ at 1.5 V, which is 12.22, 1.36 and 8.21 times higher than those of bulk NiFe-BTC (9.138 A g$^{-1}_{Ni,Fe}$), commercial Ir/C//G (82.168 A g$^{-1}_{Ir}$), and RuO$_2$//G (13.603 A g$^{-1}_{Ru}$) (Supplementary Table 6).

### Atomic modification and electronic interaction

Deconvoluted high-resolution XPS Ni 2$p$ spectrum of NiFe-BTC//G shows two peaks of Ni 2$p_{3/2}$ and Ni 2$p_{1/2}$ located at 855.7 and 873.4 eV with two satellite peaks at 861.0 and 878.4 eV, respectively, revealing the characteristic features of Ni$^{2+}$ (Supplementary Figs. 23–25)[44,45]. The two peaks of Fe 2$p_{3/2}$ at 712.0 eV and Fe 2$p_{1/2}$ at 725.9 eV accompanied by two satellite peaks at 718.5 and 734.2 eV correspond to Fe$^{3+}$ and Fe$^{2+}$, respectively, suggesting the partially reduced of Fe$^{3+}$ to Fe$^{2+}$ during the electrochemical synthetic process (Supplementary Fig. 26)[26,46]. Compared with bulk NiFe-BTC, the binding energies of Ni and Fe XPS spectra in NiFe-BTC//G have slight negative and positive shifts of -0.4 and 0.3 eV, respectively, which is a clear sign of enhanced electron transfer between Fe and Ni species under nanoconfinement from graphene multilayers. There are no obvious changes in the binding energies of Ni

and Fe XPS spectra before and after OER test, which further confirms the structural stability of NiFe-BTC//G (Supplementary Figs. 25, 26).

We then use X-ray absorption near-edge structure (XANES) and extended X-ray absorption fine structure (EXAFS) to clarify the local atomic coordination environment and electronic structure of NiFe-BTC//G. The Ni K-edge XANES of NiFe-BTC//G between Ni foil and NiO indicates the partial oxidation state of Ni species (Fig. 3a)[47,48]. The strong peak shown in Ni K-edge EXAFS spectrum of NiFe-BTC//G at 1.66 Å is mainly attributed to the scattering of Ni-O coordination, while the undetectable scattering peak related to Ni-Ni coordination demonstrates the formation of MOF structure (Fig. 3b)[14,45]. The quantitative EXAFS curve fitting analyses reveal the coordination of Ni center with six O atoms (Fig. 3c and Supplementary Table 8)[49]. The Fe K-edge XANES indicates the oxidation state of Fe in NiFe-BTC//G is +3 (Fig. 3d)[46,50,51]. The peak of 1.53 Å in Fe K-edge EXAFS spectrum of NiFe-BTC//G reveals the scattering of Fe-O coordination (Fig. 3e)[52,53]. The coordination configuration of the Fe atom in NiFe-BTC//G was further examined by quantitative EXAFS curve fitting analyses, which reveal the coordination numbers (-5.0) of Fe-O (Fig. 3f and Supplementary Table 8). The XANES results thus confirm the electron transfer between Fe and Ni in NiFe-BTC//G, which agrees to XPS results.

We further carried out the Fourier transform (FT) EXAFS fittings in R, q, and K-spaces to evaluate the structural parameters (Supplementary Figs. 27–29 and Supplementary Table 8). The wavelet transform (WT) contour plots of Ni K-edge and Fe K-edge (Supplementary Figs. 30, 31) also confirm the coordination number values of Ni-O and Fe-O in NiFe-BTC//G, which results in the form of NiO$_6$-FeO$_5$. The unsaturated coordination of NiFe-BTC//G is thus expected more reactive than the bulk NiFe-BTC with normal Fe-O and Ni-O coordinations[49,54].

### Insight into the underlying mechanism

We finally used density functional theory (DFT) calculations to study mechanistic details and active sites of the NiFe-BTC//G electrode (Fig. 4a). The projected density of state (PDOS) shows a lower energy level and larger amount of electronic resonance between Ni/Fe 3d and relevant bound O 2p orbitals for NiFe-BTC//G, compared to bulk NiFe-BTC. It indicates a stronger binding of Ni/Fe-O bonds with

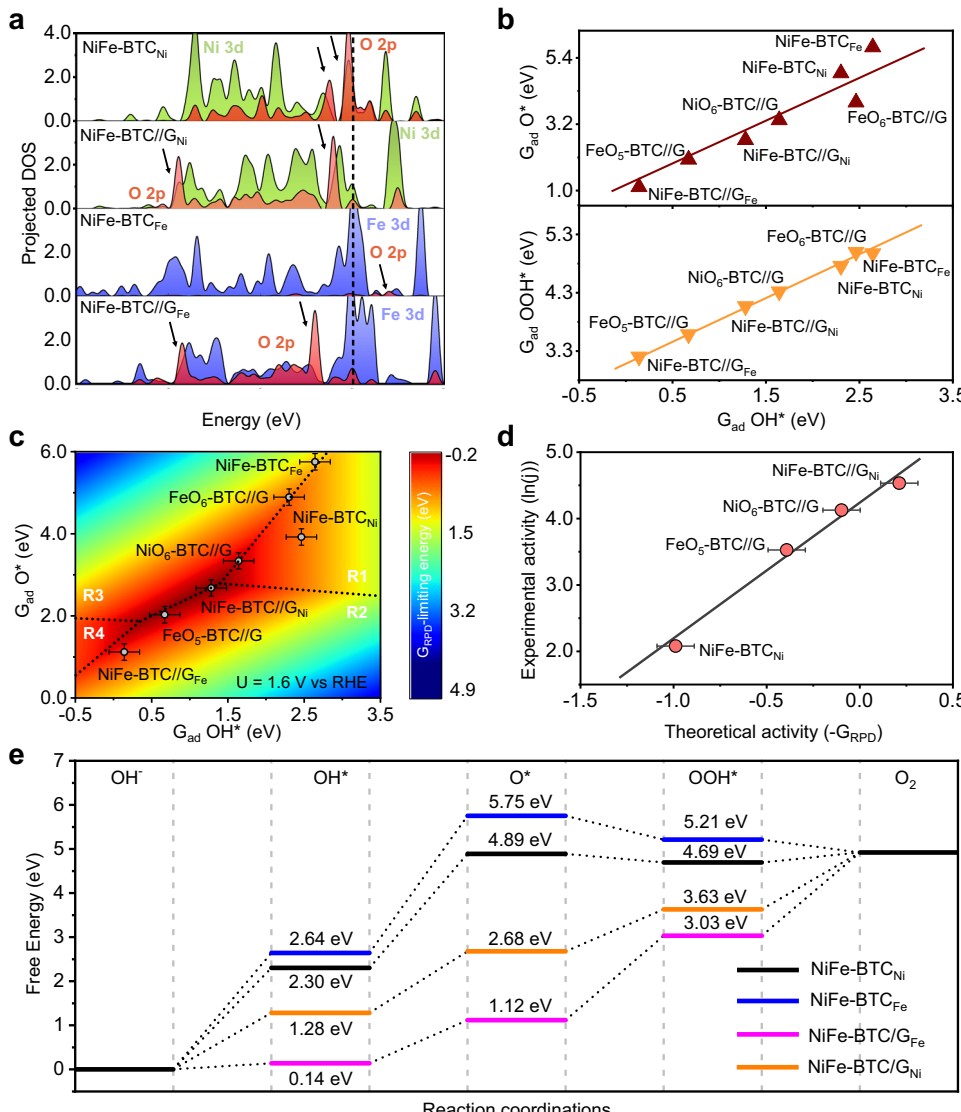

**Fig. 4 | DFT calculations of OER activity. a** PDOS of Ni/Fe 3d and O 2p orbitals for NiFe-BTC (NiFe-BTC$_{Ni}$ and NiFe-BTC$_{Fe}$ for O binding with Ni and Fe atom) and Confined NiFe-BTC (NiFe-BTC//G$_{Ni}$ and NiFe-BTC//G$_{Fe}$ for O binding with Ni and Fe atom in the confinement environment). **b** Linear scaling relationship between adsorption energies of O*/OOH* vs. OH*. All the energies have been corrected to free energy with also solvent effect considered (see all optimized structures in Supplementary Fig. 43). **c** Two-dimensional activity map with O* and OH* as two independent descriptors. The dotted lines separate the reaction phases with different limiting steps as R1, R2, R3 and R4 (refer to Supplementary Equations 4–7). The color bar shows the G$_{RPD}$-limiting energy at the electrode potential of 1.6 V vs RHE. A 0.2 eV error bar was applied due to the uncertainty of G$_{RPD}$ from the scaled adsorption energy of OOH*. **d** The correlation between experimental activities (ln($j$)) and theoretical ones (-G$_{RPD}$) derived from the G$_{RPD}$-limiting energies. A 0.2 eV error bar was applied due to the uncertainty of G$_{RPD}$ from the scaled adsorption energy of OOH*. **e** Free energy diagram of OER on NiFe-BTC and NiFe-BTC//G with both active sites of Ni and Fe (see free energy diagrams of OER on other sites in Supplementary Fig. 34).

nanoconfinement effect. Note that, a quite difficult O* formation, namely a weak O* binding energy, was reported in several previous works for Ni/Fe sites with coordinated O[55–57]. According to the theoretical OER volcano trend, a promotion of OER activity can be achieved with the strengthening of Ni/Fe-O bond, which was expected to be performed by the confinement effect in this work.

Accordingly, we calculated the adsorption energies of O*, OH*, and OOH* on different MOF structures, considering both Fe and Ni sites (Supplementary Fig. 32 and Supplementary Table 9). The correlations between the adsorption-free energies of O*/OOH* and OH* (descriptor) were established, namely the scaling relations (Fig. 4b), where a two-dimensional linear fitting (Supplementary Fig. 33) was further built towards a more accurately description of OOH* adsorption energies scaled by both O* and OH* adsorption energies. The four steps of proton-coupled electron transfer were applied for OER in

alkaline conditions (Supplementary Equations 4–7). Hereafter, a two-dimensional (2D) activity map was built at the electrode potential of 1.6 V vs RHE, namely the reaction phase diagram (RPD) (Fig. 4c).

According to the 2D activity map, higher OER activity, described by the limiting energies (G$_{RPD}$-limiting energies; Supplementary Equation 13), was found on confined MOF structures rather than bulk ones. More importantly, a superior activity can be found for OER on NiFe-BTC//G with the active Ni site (NiFe-BTC//G$_{Ni}$), rather than NiO$_6$-BTC//G, due to the stronger binding of Ni-O bond from the synergy of Ni and Fe. However, the too strong Fe−O bond results in a more difficult OH* deprotonation (Supplementary Equation 5), displaying lower OER activity on NiFe-BTC//G$_{Fe}$. On the other hand, the 2D activity map shows an activity trend of OER to be NiFe-BTC//G$_{Ni}$ > NiO$_6$-BTC//G > FeO$_5$-BTC//G > bulk NiFe-BTC. The great agreement (Fig. 4d) between the theoretical and experimental activities (Fig. 2a) confirms

the reliability of the 2D activity map. Finally, the free energy diagram of OER on NiFe-BTC//$G_{Ni}$ together with that on NiFe-BTC//$G_{Fe}$, NiFe-BTC$_{Ni}$, and NiFe-BTC$_{Fe}$ indicates again the lower limiting potential on confined structures (Fig. 4e, see all other free energy diagrams in Supplementary Fig. 26), where the potential-limiting step is the OH* deprotonation (1.40 eV at 0 V vs. RHE, namely an overpotential of 0.17 V) on NiFe-BTC//$G_{Ni}$. It confirms again the lower overpotential from the experiment for NiFe-BTC//G (106 mV) rather than that on Ni-BTC//G (212 mV) Fe-BTC//G (226 mV) (see detailed data in Supplementary Figs. 35–39, and Supplementary Table 3).

## Discussion

In summary, we have developed an electrochemical strategy to endow poorly conductive MOFs with enhanced catalytic performance through the nanoconfinement by graphene multilayers. Such nanoconfinement optimizes the electronic structure and catalysis center of MOF materials as well as lowers the limiting potential for electrochemical reactions. Therefore, the as-prepared NiFe-BTC//G-2h shows a rather low OER overpotential of 106 mV to reach 10 mA cm$^{-2}$ with a good stability of over 150 h. Our work challenges the common conception of pristine MOFs as inert catalysts and sheds light on utilizing less conductive or even insulating MOFs into electrocatalytic applications.

## Methods

### Electrode preparation

**Synthesis of NiFe-BTC//G-2h.** The commercial graphite foil was successively rinsed with acetone, ethanol, and DI-water under sonication for 20 min, and then dried in oven at 60 °C for 2 h. A simple two-electrode system was employed for intercalation of graphite with ions and organic molecules, in which platinum foil and the commercial graphite foil were kept in parallel at a constant distance of 2.5 cm and placed as cathode and anode, respectively. Firstly, an electrochemical exfoliation process was conducted by anodization of graphite foil in dilute sulfuric acid solution (0.5 M, 50 mL) with using Pt foil (10 × 10 mm) as counter electrode and graphite foil (10 × 30 mm) as working electrode under 3 V for 30 min. The exfoliated graphite foil was rinsed with DI-water several times and dried in oven at 60 °C. Secondly, 420 mg of 1,3,5-benzenetricarboxylic acid and 240 mg of NaOH were dissolved in DI-water (75 mL) and rigorously stirred for 30 min to obtain transparent solution. Then, the dilated graphite foil was put into the above solution and intercalated by 1,3,5-benzenetricarboxylic group under 5 V for 2 h. The as-obtained graphite foil was immersed into DI-water and slightly sonicated at an ultrasonic frequency of 20 KHz for 30 min to remove the excessive organic 1,3,5-benzenetricarboxylic trisodium salt on the surface of graphite foil as well as small pieces of deciduous graphite flakes. Finally, 713 mg of NiCl$_2$ • 6H$_2$O and 1.21 g of Fe(NO$_3$)$_3$ • 9H$_2$O were dissolved in a mixed solvent of DI-water and ethanol (100 mL, V:V = 1:1) and stirred for 20 min to form transparent inorganic salt solution. The treated graphite was immersed into the above solution for 24 h, during which the NiFe-BTC was in situ fabricated. The as-prepared electrode was then taken out, rinsed with copious DI-water and ethanol, dried at 60 °C overnight, and directly employed to trigger the anodic reaction of water splitting without carbonization treatment.

### Characterizations

The morphologies of samples were examined by transmission electron microscopy (TEM, HT7700), and high-resolution transmission electron microscopy (HRTEM, JEOL JEM-2001F). The chemical environments of samples were measured by X-ray photoelectron spectroscopy (XPS) (Escalab 250Xi) with Al Kα radiation. Raman spectra of samples were tested with a LabRAM HR Evolution. The N$_2$ adsorption-desorption curves of samples were measured by Brunauer-Emmett-Teller (BET) (ASIC-2). The metal content in catalysts was investigated by ICP-MS (Vista Axial).

### Electrochemical measurements

All electrochemical measurements were carried out by an electrochemical analyzer (CHI 760E) in a typical three-electrode configuration. The Ag/AgCl (saturated KCl) with a salt bridge and Hg/HgO were used as the reference electrodes, while a graphite rod was employed as the counter electrode. The potential was converted to reversible hydrogen electrode (RHE) via a Nernst equation ($E_{RHE} = E_{Ag/AgCl} + 0.059 \times pH + 0.197$; $E_{RHE} = E_{Hg/HgO} + 0.059 \times pH + 0.098$). To investigate oxygen evolution reaction (OER) performances of electrocatalysts, the scan rate of linear sweep voltammetry (LSV) was set as 1.0 mV s$^{-1}$ with the potentials between 0 V and 0.8 V vs. Ag/AgCl or Hg/HgO in 1.0 M KOH. Electrochemical impedance spectroscopy (EIS) was measured at 0.5 V vs. Ag/AgCl with a frequency range from 105 to 0.01 Hz. All polarization curves were calibrated without iR correction unless noted. Cyclic voltammetry cycles (CVs) at 0.8–1.0 V vs. RHE with the scan rates from 20 to 100 mV s$^{-1}$ was applied to analyze electrochemically active surface area (ECSA).

### Computational details

Density functional theory (DFT) calculations were carried out through the Vienna ab initio simulation package (VASP)[58,59]. The revised Perdew-Burke-Ernzerhof (rPBE) functional[60] was applied with the basis set of plane-wave by the method of the Projector-augmented wave (PAW)[61,62]. The cut-off energy was set to be 400 eV, where the reliability of accuracy was confirmed with the effect <0.02 eV (Supplementary Table 10). The spin magnetization was tested to show less effect on the system (Supplementary Table 12). The structure of bulk single-Ni/Fe metal organic frameworks (MOFs) was constructed by Ni/Fe atom as the central site, where two 1,3,5-benzenetricarboxylate groups and four hydroxyl groups were applied as the ligand for six coordinated O groups in total. The structure of bulk NiFe-BTC was built by two metal atoms, where two 1,3,5-benzenetricarboxylate groups, two O atoms and six hydroxyl groups were applied as the ligand for six coordinated O groups for Ni and Fe. Two layers of graphene were built to describe the confinement environment (Supplementary Fig. 32). Different from the bulk NiFe-BTC that generally consisted of NiO$_6$ and FeO$_6$ sites, the metallic ions (Ni and Fe) of confined NiFe-BTC (described as NiFe-BTC//G) are coordinated into an octahedron, where Ni and Fe interconnect with six and five coordinated O/hydroxyl groups respectively, as indicated by extended X-ray absorption fine structure (EXAFS, NiO$_6$-FeO$_5$ units) (Supplementary Fig. 32). Note that, one NiFe (also for Ni or Fe) unit with two BTC structures were constructed for all the calculations, where the effect from 3D structures were tested to show less effect on the calculated energies (Supplementary Fig. 43, Supplementary Table 13). Monkhorst-Pack k-points of 1 × 1 × 1 was applied to all the calculations due to the large size of the unit cell. The convergence of force was set to 0.05 eV Å$^{-1}$, which was also tested to be accurate enough with the energy effect <0.02 eV based on more rigorous force convergence (Supplementary Table 10).

## Data availability

The data that support the findings of this study are available from the corresponding authors upon reasonable request. Source data are provided as a Source data file. Source data are provided with this paper.

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

## Acknowledgements

Y.H. acknowledges the financial support from the National Natural Science Foundation of China (22278364, 22211530045, 22178308, 21922811, 21878270), the Zhejiang Provincial Natural Science Foundation of China (LR19B060002), the Startup Foundation for Hundred-Talent Program of Zhejiang University. T.Z. acknowledges the Excellent Youth Foundation of Zhejiang Province of China (Grant No. LR21E030001), the National Natural Science Foundation of China (Grant No. 52003279), and Science Foundation of Donghai Laboratory (Grant No. DH-2022KF0310). Y.H and T.Z. acknowledge the financial support from the Key Laboratory of Marine Materials and Related Technologies (CAS, Grant No. 2020K10). J.X. acknowledges the financial support from the National Key Research and Development Program of China (No. 2021YFA1500702), DNL cooperation Fund, CAS (DNL202003), National Natural Science Foundation of China (Nos. 22172156 and 91945302), Strategic Priority Research Program of the Chinese Academy of Sciences (XDB36030200). XAS data were collected at BL14W1 beamline in Shanghai Synchrotron Radiation Facility (SSRF), Beijing Synchrotron Radiation Facility (BSRF), and Hefei Synchrotron Radiation Facility (HSRF).

## Author contributions

Y.H., T.Z., and J.X. conceived the idea and supervised the project. S.L. performed electrolysis experiments and measurements, and carried out data analysis. J.W. performed the synthesis of MOFs. C.G. implemented the density-functional theory calculations. Z.L., B.Y., and L.L. supervised the electrolysis measurements. L.W. asissted in the material characterization and discussion. S.L. and C.G. wrote the manuscript. All authors discussed the results and assisted with manuscript preparation.

## Competing interests

The authors declare no competing interests.
