## [Peer Review File · Nature Communications]

Exceptional Catalytic Activity of Oxygen Evolution Reaction via Two-Dimensional Graphene Multilayer Confined Metal-Organic FrameworksREVIEWER COMMENTS

Reviewer #1 (Remarks to the Author):

The manuscript reported a new method to prepare efficient electrocatalytic materials for oxygen evolution reactions. The authors introduced BTC ligands and Fe and Ni ions (from their salt solutions) in-between the graphene layers with electrochemical synthesis. The synthesized materials show impressive electrocatalytic properties for oxygen evolution from water with very low overpotential (106mV). The material was well characterized, and the mechanism was studied and proposed in the manuscript. The topic has a broad interest in materials society and the publication is strongly recommended. The following are some suggestions for consideration during revision:

The carboxylic group from BTC located between the graphene layer may coordinate with metal ions as reported here. In addition, can authors comment on the possibilities of OH⁻ from graphene surface to coordinate with metal ions? Can the authors estimate the distances between the graphene layer to see if a single layer of metal-BTC with OH⁻ or water fit into the space?

Reviewer #2 (Remarks to the Author):

Lyu et al report on the formation of Graphene multilayer confined metal-organic frameworks and their exceptional catalytic activity toward oxygen evolution. The authors presented both material synthesis and characterization in a very detailed manner. The interpretation of materials characterization is also consistent. In contrast, the results and interpretation of the electrochemical characterization are questionable.

First of all, the authors used a Ag/AgCl reference electrode that is not supposed to be used in alkaline media. Studies showed that the Ag/AgCl reference electrode is not stable in alkaline media [DOI: 10.1617/s11527-015-0673-8]. In addition, in alkaline media, AgOH forms due to the reaction between Ag and hydroxyl ion, which quickly turns into Ag₂O (black ppt.) and blocks the membrane. Therefore, the potential measurements in this study may not be correct.

Another issue in the electrochemical results is the unfair comparison between synthesized catalysts and Ir/C and RuO₂. In electrocatalysis, comparing different materials requires fixing the particle size to determine/compare the intrinsic activity of the various materials. In this study, the authors compared Ir/C to RuO₂ without providing any information about the catalyst loading used or the morphology of the catalysts. In addition, an oxide material (RuO₂) was compared to a metallic catalyst (Ir), which is not a fair comparison as Ir will oxidize during testing and form a hydrous oxide, which will in turn provides very high activity. It is known that Ru(RuO₂) catalysts are more active than IR(IrO₂), but according to the results shown here, the Ir/C is more active than RuO₂, which can be due to one or more of the following:

1. A significant difference in the catalyst loading on the electrode
2. Ir nanoparticles in Ir/C are much smaller than RuO₂
3. Metallic Ir converts to hydrous Ir oxide, while RuO₂ is not doing so
4. carbon corrosion in Ir/C contributes significantly in the measured current

To conclude this point, if the loadings of Ir/C and RuO₂ are not known, how should their activities be compared to the investigated MOF/G-based catalysts?

The authors performed the measurements on stagnant electrodes, i.e., no rotation, while it became well established that bubbles and microscopic bubbles can block the active sites of the catalysts, even during rotations [DOI: 10.1149/1945-7111/abdcc9, DOI: 10.1021/acsaem.0c01944, and DOI: 10.1149/2.0301908jes]. Therefore, for stagnant electrodes, it is expected that the effect of bubbles

will be much worse. Is it then possible that the extent of bubbles sticking and shielding of active sites is higher for Ir/C and RuO₂ compared to MOF/G-based catalysts? If this is correct, then the MOF/G-based catalysts will appear much more active.

The authors are advised to use an RDE technique or a similar technique to decrease the effect of bubbles accumulation.

In supplementary Figures 5, 6, and 12, the onset potential for the OER is almost at the thermodynamic potential, 1.23 V, is this even possible? Can this be due to the wrong Ag/AgCl reference potential as indicated above?

Reviewer #3 (Remarks to the Author):

The article from Siliu Lyu et al. shows a novel strategy to increase the OER activity of MOFs by intercalating them within graphene layers. This allows achieving low overpotential and long stability of the resulting electrode. The study employs different microscopic, electrochemical and spectroscopic techniques to fully characterize the NiFe-MOF//G electrode. Furthermore, the use of Ni and Fe XANES and EXAFS hints about the coordination environment of the metallic cations in the MOF structures and DFT calculations with periodic boundary conditions are employed to shed light on the reaction mechanism. The strategy is broadened to several MOF structures, although better performance is obtained when using BTC as organic precursor.

The assignment of the EXAFS spectra (Figs 3b and e) should include the fits of different scattering paths and not only the comparison with reference samples. Moreover, after analyzing the XAS results, the authors propose a dimeric NiFe structure (Figure 3f). Given that this structure is key for the following DFT calculations and reaction mechanism analysis, it would be highly enlightening if the authors include more information as to how they reached to that coordination configuration.

As to the DFT calculations, it would be good practice to include in the supplementary document input files and optimized geometries employed. The authors should also explain their choice of cut-off energy and Monkhorst-Pack grid, either with convergence test calculations or by citing adequate literature.

The study focuses on the MOF formed by BTC but also presents results with different MOF structures that show worse performance (Supplementary Fig. 13). It would be important that the authors give us some explanation about the higher overpotentials observed with MOFs formed with other organic precursors than BTC. Did the authors perform DFT calculations with the other MOFs to account for the experimental results obtained?

The experimental section explains the synthesis procedure of the intercalated NiFe-MOF//G. It would be important to know what the authors refer to with "The as-obtained graphite foil was immersed into DI-water and slightly sonicated for 30 min".

Reviewer #4 (Remarks to the Author):

The authors present a detailed study of a hybrid electrocatalytic material built from graphite with intercalated organic complexes of Ni/Fe. The manuscript describes the synthesis, structural characterization via XPS, characterization of electro catalytic activity via cyclic voltammeter and chronopotentiometric testing, and computational characterization via DFT simulations. The material is found to have a significantly low over potential for the oxygen evolution reaction (OER). Overall the results seem interesting and of potential high impact. However, a few aspects of the description and of the analysis of the results appear unclear or incomplete.

* The authors refer to the material as confined MOFs and this is what Figure 1, in particular 1b would suggest. However, the later discussion identify the material as small complexes of organic ligands and metal ions, embedded within the graphite layers. I would expect MOFs to be associated with crystalline porous materials, rather than molecular-level complexes.

* Connected to the use of the MOF terminology the authors correctly point out that some of the limitations of MOFs is the poor conductivity and poor mass transport during catalysis. They also identify porosity and high surface area as one of the advantages. However, in the proposed material the structural porosity of MOFs is clearly not an issue. Mass transport within graphite sheets is not explicitly discussed, although the experimental results would suggest it is not a limiting factor for this material. If I understand correctly the strategy proposed by the authors is to select promising combinations of organic ligands and metal centers from existing MOFS and intercalate them within graphite.

* While the experiments seem to suggest that this is a successful strategy, as the complexes in exam become more active, a rigorous explanation on why the structures are more reactive is not present. From a computational perspective, inclusion of solvent molecules in the simulations is probably very relevant. The authors adopt continuum embedding as a correction, which I assume is computed on more ideal structures of intermediates and added a posteriori to each specific simulation. This is a significant simplification on top of an already simplified approach. Moreover, an accurate description of the effect of confinement from graphite would strongly depend on the positions of the first or second solvation layer. While some of the computational details are missing or not properly justified (revPBE may not be the most appropriate functional, how is spin magnetization handled, is the graphite layer allowed to relax, etc.), it is true that they may be less relevant in a comparative analysis. However, appropriate accounting of hydrogen bonds and statistical sampling may significantly alter the computed free energy diagrams. Moreover, it would be useful to understand if the structure of the confined solvent contributes to the enhanced reactivity and in which way it affects mass transport within the graphite layers.

* The structures of the simulated intermediates should be included in the supplementary information, as they would allow to evaluate the type of catalytic pathways that have been explicitly analyzed. My understanding of the computational details suggests that the authors are using the Computational Hydrogen Electrode approach to compute the effect of the potential on the catalytic activity, references to the appropriate literature would be important. It is also not clear if the calculations reported for the non-confined MOFs are computed for bulk 3D materials or for small molecular complexes without the graphite layers (as visualized in Figure 24)? As a very minor note, in the Computational Details of the main manuscript the authors refer twice to supplementary Figure 23 instead of 24.

Provided the points above are fully addressed, I would recommend publication of the manuscript on Nature Communications.

Responses for the Nature Communications

Manuscript ID: NCOMMS-22-09218-T

Title: Exceptional Catalytic Activity of Oxygen Evolution Reaction via Two-Dimensional Graphene Multilayer Confined Metal-Organic Frameworks

(The explanations to the comments from reviewers are shown in blue color. The modifications provided in the revised manuscript and Supplementary Information are shown in blue color with yellow highlight.)

Responses to the comments of reviewers:

Reviewer #1 (Remarks to the Author):

The manuscript reported a new method to prepare efficient electrocatalytic materials for oxygen evolution reactions. The authors introduced BTC ligands and Fe and Ni ions (from their salt solutions) in-between the graphene layers with electrochemical synthesis. The synthesized materials show impressive electrocatalytic properties for oxygen evolution from water with very low overpotential (106 mV). The material was well characterized, and the mechanism was studied and proposed in the manuscript. The topic has a broad interest in materials society and the publication is strongly recommended.

Response: We appreciate the reviewer for the valuable comments on our manuscript.

The following are some suggestions for consideration during revision:

The carboxylic group from BTC located between the graphene layer may coordinate with metal ions as reported here. In addition, can authors comment on the possibilities of OH⁻ from graphene surface to coordinate with metal ions?

Response: We are thankful to the reviewer for the professional comment. We totally agree the reviewer that the carboxylic group from oxidized graphene could be possible to coordinate with Ni and Fe ions, but the effect of the functional groups (i.e., OH⁻, COOH⁻) from graphene surface on the OER activity should be minor. Regarding the reviewer's concern, we did additional experiments to verify our assumption. Firstly, we prepared the expanded graphite foil through electrochemical exfoliation process in dilute sulfuric acid solution (0.5 M, 50 mL). Secondly, the expanded graphite foil was directly immersed in Ni and Fe salt solution without the introduction of organic ligands (denoted as NiFe//G). Their OER performances are compared through an electrochemical analyzer (CHI 760E) in a typical three-electrode configuration. As shown in Fig. R1, the NiFe//G has an averagely lower overpotential of 338 mV to achieve the current density of 10 mA cm⁻² compared to the graphite foil (376 mV at 10 mA cm⁻²). However, their overpotentials are both much higher than that of NiFe-

BTC//G (106 mV at 10 mA cm⁻²), which demonstrates that the effect of active group on graphene surface could be neglected.

Fig. R1 | (a) LSV plots obtained with NiFe//G and graphite foil for OER at 10 mV s⁻¹ in 1.0 M KOH. (b) Comparison of the required voltages at 10 mA cm⁻² for NiFe//G and graphite foil.

The above results have been included in the Supplementary Fig. 6 in the revised Supplementary Information.

Can the authors estimate the distances between the graphene layer to see if a single layer of metal-BTC with OH⁻ or water fit into the space?

Response: We thank the reviewer for the thoughtful comment. We would like to note that the distance of graphene layers after electrochemical intercalation is actually in a wide range from a few angstrom (as viewed by HRTEM, Supplementary Fig. 3) and a few nanometers and even tens of micrometers (as viewed by SEM, Fig. R2). As such, both the MOF and water could fit into the space between graphene layers.

Fig. R2 | SEM and element mapping images of NiFe-BTC//G-2h electrode. Scale bars, 2.0 μm.

The results of above SEM and element mapping images have been included in the Supplementary Fig. 2 in the revised Supplementary Information.

Reviewer #2 (Remarks to the Author):

Lyu et al report on the formation of Graphene multilayer confined metal-organic frameworks and their exceptional catalytic activity toward oxygen evolution. The authors presented both material synthesis and characterization in a very detailed manner. The interpretation of materials characterization is also consistent. In contrast, the results and interpretation of the electrochemical characterization are questionable.

Response: We appreciate the reviewer for the valuable comments on our manuscript. The electrochemical results and discussions of the manuscript has been modified according to your suggestions.

First of all, the authors used a Ag/AgCl reference electrode that is not supposed to be used in alkaline media. Studies showed that the Ag/AgCl reference electrode is not stable in alkaline media [DOI: 10.1617/s11527-015-0673-8]. In addition, in alkaline media, AgOH forms due to the reaction between Ag and hydroxyl ion, which quickly turns into Ag₂O (black ppt.) and blocks the membrane. Therefore, the potential measurements in this study may not be correct.

Response: We thank the reviewer for the comments. In our work, the Ag/AgCl reference electrode was protected by the salt bridge (Fig. R3). The Ag/AgCl reference electrode contacted directly with the saturated KCl solution rather than the alkaline solution. We note that the Ag/AgCl reference electrode protected by the salt bridge was often used in OER test for alkaline solution (Nat Catal., 2022, 5: 414-429; Nat Catal., 2021, 4: 212-222; Nat Commun., 2022, 13: 2191; J. Am. Chem. Soc., 2022, 144: 9254-9263).

Fig. R3 | Digital photographs of (a) Ag/AgCl electrode, (b) salt bridge filled with saturated KCl solution, and (c) the Ag/AgCl electrode used in this work.

Regarding the reviewer's concern, we performed additional experiments to investigate the stability of Ag/AgCl (saturated KCl) during the OER test. We use the NiFe-BTC//G and noble metal based electrocatalysts directly as electrodes to catalyze the anodic reaction of water splitting in alkaline solution via a typical three-electrode system with the Ag/AgCl (saturated KCl) and Hg/HgO as reference electrodes for comparison. To further discuss the rationality of our results, the noble metal based Ir/C and RuO₂ electrocatalysts were prepared by using same substrate of NiFe-BTC//G-2h for comparison (denoted as Ir/C//G and RuO₂//G, respectively). The NiFe-BTC//G displayed similar overpotentials when the Ag/AgCl (~118 mV) and Hg/HgO (~130 mV) were worked as reference electrodes, respectively (Fig. R4). The Ag/AgCl (saturated KCl) electrode was proved to be stable in alkaline solution when protected by the salt bridge, which can be attributed to the directly contact of Ag/AgCl with the saturated KCl solution rather than the alkaline solution. On the other hand, the chronopotentiometry measurement for the NiFe-BTC//G displayed an insignificant potential change over 150 h of continuous reaction at a constant current density of 10 mA cm⁻² in 1.0 M KOH when employing Ag/AgCl (saturated KCl) electrode as reference (Fig. 2d), which further confirmed the relative stability of Ag/AgCl (saturated KCl) electrode in alkaline solution. Similarly, the Ir/C//G displayed similar overpotentials when the Ag/AgCl (~287 mV) and Hg/HgO (~292 mV) were worked as reference electrodes, respectively (Fig. R5a). Also, the RuO₂//G showed comparable overpotentials of 267 and 264 mV with Ag/AgCl (saturated KCl) and Hg/HgO as reference electrodes, respectively (Fig. R5b). The OER performances of noble metal based electrocatalysts further proved the stability of Ag/AgCl (saturated KCl) electrode in 1.0 M KOH.

Fig. R4 | LSV plots obtained with NiFe-BTC//G for OER at 10 mV s⁻¹ in 1.0 M KOH. Red: Ag/AgCl

(saturated KCl) as the reference electrode; Blue: Hg/HgO as the reference electrode. Inset: A close-up of LSV plots.

Fig. R5 | LSV plots obtained with (a) Ir/C//G and (b) RuO₂//G for OER at 10 mV s⁻¹ in 1.0 M KOH. Red: Ag/AgCl (saturated KCl) as the reference electrode; Blue: Hg/HgO as the reference electrode. Inset: Close-ups of LSV plots.

The above discussion and electrode preparation have been included in the revised Supplementary Information and Supplementary Figs. 7 and 8.

Another issue in the electrochemical results is the unfair comparison between synthesized catalysts and Ir/C and RuO₂. In electrocatalysis, comparing different materials requires fixing the particle size to determine/compare the intrinsic activity of the various materials. In this study, the authors compared Ir/C to RuO₂ without providing any information about the catalyst loading used or the morphology of the catalysts. In addition, an oxide material (RuO₂) was compared to a metallic catalyst (Ir), which is not a fair comparison as Ir will oxidize during testing and form a hydrous oxide, which will in turn provides very high activity. It is known that Ru(RuO₂) catalysts are more active than IR(IrO₂), but according to the results shown here, the Ir/C is more active than RuO₂, which can be due to one or more of the following:

1. A significant difference in the catalyst loading on the electrode
2. Ir nanoparticles in Ir/C are much smaller than RuO₂
3. Metallic Ir converts to hydrous Ir oxide, while RuO₂ is not doing so
4. carbon corrosion in Ir/C contributes significantly in the measured current

To conclude this point, if the loadings of Ir/C and RuO₂ are not known, how should their activities be compared to the investigated MOF/G-based catalysts?

Response: We appreciate the thoughtful comments from the reviewer. In the manuscript, the loading amount of bulk NiFe-BTC was determined to 1.0 mg cm⁻². For the optimized NiFe-BTC//G, the loading amount was normalized to 7.3 mg cm⁻² by the weight difference method for 20 times. Besides, to accurately evaluate the activities of various electrocatalysts, the commercial Ir/C and RuO₂ were further combined with the

substrate of expanded graphite foil, denoted as Ir/C//G and RuO₂//G, respectively. Both the Ir/C//G and RuO₂//G have the loading amount of 1.0 mg cm⁻².

Regarding the reviewer's concern for the comparison of activities, we adopted the metal mass activity to compare the OER performance of various prepared catalysts (Adv. Mater., 2020, 32: 2001430; Nat Catal., 2018, 1: 841-851; Nat Commun., 2019, 10: 162).

Considering the different loading amount of NiFe-BTC//G, bulk NiFe-BTC, Ir/C//G, and RuO₂//G, the metal mass activity at 1.5 V vs. RHE was adopted to compare their OER performance. As shown in Fig. 2, the NiFe-BTC//G reaches the highest mass activity of 111.643 A g_{Ni,Fe}⁻¹ at 1.5 V vs. RHE, which is 12.22, 1.36 and 8.21 times higher than those of bulk NiFe-BTC (9.138 A g_{Ni,Fe}⁻¹), commercial Ir/C//G (82.168 A g_{Ir}⁻¹), and RuO₂//G (13.603 A g_{Ru}⁻¹) (Table R1), respectively. Among them, the Ir/C//G obtains the metal mass activity of 82.168 A g_{Ir}⁻¹ at 1.5 V vs. RHE, which is higher than that of RuO₂//G (13.603 A g_{Ru}⁻¹). This could be attributed to the promotional effect of expanded graphite foil on Ir as well as the conversion of metallic Ir to hydrous Ir oxide.

The corresponding revisions and discussions were described in the revised manuscript (Page 4-6). Furthermore, Fig. 2 was replaced by Fig. R6 in the main text, and the related tables were added in the revised Supplementary Information (Supplementary Tables 3 and 6).

Fig. R6 OER electrochemical activity of NiFe-BTC//G. (a) LSV plots obtained with NiFe-BTC//G, bulk NiFe-BTC, commercial Ir/C//G and RuO₂//G for OER at 10 mV s⁻¹ in 1.0 M KOH. (b) Tafel plots obtained with NiFe-BTC//G, bulk NiFe-BTC, commercial Ir/C//G and RuO₂//G. (c) EIS Nyquist plots. (d) Chronopotentiometric testing of NiFe-BTC//G for 150 h at 10 mA cm⁻² in 0.1 M KOH. (e) Comparison of the required voltage at 10 mA cm⁻² for NiFe-BTC//G with other state-of-the-art MOF-based electrocatalysts.

Table R1. The metal mass activities at 1.5 V of various samples.

Samples	Loading amount (mg cm ⁻²)	Metal mass activity at 1.5 V
NiFe-BTC//G-2h	13.82	111.643 A g _{Ni,Fe} ⁻¹
Bulk NiFe-BTC powder	1.0	9.138 A g _{Ni,Fe} ⁻¹
Ir/C//G	1.0	82.168 A g _{Ir} ⁻¹
RuO ₂ //G	1.0	13.603 A g _{Ru} ⁻¹

The authors performed the measurements on stagnant electrodes, i.e., no rotation, while it became well established that bubbles and microscopic bubbles can block the active sites of the catalysts, even during rotations [DOI: 10.1149/1945-7111/abdcc9, DOI: 10.1021/acsaem.0c01944, and DOI: 10.1149/2.0301908jes]. Therefore, for stagnant electrodes, it is expected that the effect of bubbles will be much worse. Is it then possible that the extent of bubbles sticking and shielding of active sites is higher for Ir/C and RuO₂ compared to MOF/G-based catalysts? If this is correct, then the MOF/G-based catalysts will appear much more active.

The authors are advised to use an RDE technique or a similar technique to decrease the effect of bubbles accumulation.

Response: We are thankful to the reviewer for the valuable suggestion. We totally agree the reviewer that bubbles and microscopic bubbles would block the active sites of the catalysts. To explore the influences of traditional electrode during rotations, we firstly investigated the OER performance of NiFe-BTC//G, RuO₂//G and Ir/C//G in a typical three-electrode configuration under different magnetic stirring from 0 to 2400 rpm (Fig. R7) with Hg/HgO as a reference electrode. Secondly, we adopted the suggestion from the reviewer to use an RDE technique to investigate the OER performance of NiFe-BTC//G, RuO₂//G and Ir/C//G (Fig. R8 and R9).

Fig. R7 | Overpotentials of (a) NiFe-BTC//G, (b) Ir/C//G, and (c) RuO₂//G tested under rotation at the rotate speed from 0 to 2400 rpm for OER at 10 mV s⁻¹ in 1.0 M KOH using Hg/HgO as the reference electrode. (d) LSV plots obtained with NiFe-BTC//G, Ir/C//G, and RuO₂//G under rotation for OER at

10 mV s⁻¹ in 1.0 M KOH using Hg/HgO as the reference electrode.

As shown in Fig. R7a-c, the overpotentials of NiFe-BTC//G, Ir/C//G, and RuO₂//G relocate with the rotate speed, which are within ranges of 130-151 mV, 292-334 mV, and 264-285 mV respectively. The NiFe-BTC//G displays the smallest overpotential of ~130 mV at 10 mA cm⁻² at the rotate speed of 800 rpm. While the Ir/C//G and RuO₂//G display the overpotential of ~292 and ~264 mV at the rotate speed of 1200 rpm, respectively (Fig. R7d). Excessively low rotate speed may result in the block of metal active sites by a mass of bubbles while high rotate speed may reduce the contact time of intermediates with metal active sites, both of which eventually lead to poor OER performances.

Regarding the reviewer's concern, the rotating disk electrode (RDE) results show that the overpotentials of NiFe-BTC//G, Ir/C//G, and RuO₂//G were within ranges of 98-142 mV, 243-320 mV, and 212-276 mV respectively (Fig. R9a-c). The NiFe-BTC//G shows the lowest overpotential of ~98 mV at 10 mA cm⁻² at 1200 rpm. While the Ir/C//G and RuO₂//G show the lowest overpotential of ~243 and ~212 mV through the RDE technique at 1200 rpm, respectively (Fig. R9d). The high overpotential obtained at low rotate speed could be attributed to the generated bubbles that block the active sites. While the excessive rotate speed with large shear force would introduce undesired vortex which impedes the contact of intermediates with metal active sites

Fig. R8 | Digital photographs of RRDE-3A rotating ring disk electrode apparatus and the RDE accessory used in this work.

Fig. R9 | Overpotentials of (a) NiFe-BTC//G, (b) Ir/C//G, and (c) RuO₂//G tested through an RDE technique under rotation at the rotate speed from 0 to 2400 rpm for OER at 10 mV s^{-1} in 1.0 M KOH using Hg/HgO as the reference electrode. (d) LSV plots obtained with NiFe-BTC//G, Ir/C//G, and RuO₂//G tested through the RDE technique for OER at 10 mV s^{-1} in 1.0 M KOH using Hg/HgO as the reference electrode.

The above modification and discussion have been included in the Supplementary Figs. 9 and 10 in the revised Supplementary Information.

In supplementary Figures 5, 6, and 12, the onset potential for the OER is almost at the thermodynamic potential, 1.23 V, is this even possible? Can this be due to the wrong Ag/AgCl reference potential as indicated above?

Response: We are thankful to the reviewer for the valuable comments. In our work, the lowest onset potential of NiFe-BTC//G is identified as 1.271 V vs. RHE. Given the reviewer's concern, Faradic efficiencies at different potentials from 1.2 to 1.6 V vs. RHE were calculated to investigate the onset potential and verify whether the observed current originates from OER rather than other side reactions (Fig. R10).

As shown in Fig. R10, the O₂ Faraday efficiency for the NiFe-BTC//G is $\sim 78\%$ at 1.271 V vs. RHE, and remains $94 \pm 2\%$ from 1.3 to 1.6 V vs. RHE, further confirming that the low onset potential is from the OER electrocatalysis.

Overall, the onset potential of 1.271 V vs. RHE for the NiFe-BTC//G is reasonable. Further, the Ag/AgCl electrode (saturated KCl) used in the experiments and its reference potential have been proven correct in OER tests.

Fig. R10 | Faraday efficiencies for NiFe-BTC//G tested from 1.2 to 1.6 V vs. RHE.

The results of above Faraday efficiencies for NiFe-BTC//G have been included in the Supplementary Fig. 22 in the revised Supplementary Information.

Reviewer #3 (Remarks to the Author):

The article from Siliu Lyu et al. shows a novel strategy to increase the OER activity of MOFs by intercalating them within graphene layers. This allows achieving low overpotential and long stability of the resulting electrode. The study employs different microscopic, electrochemical and spectroscopical techniques to fully characterize the NiFe-MOF//G electrode. Furthermore, the use of Ni and Fe XANES and EXAFS hints about the coordination environment of the metallic cations in the MOF structures and DFT calculations with periodic boundary conditions are employed to shed light on the reaction mechanism. The strategy is broadened to several MOF structures, although better performance is obtained when using BTC as organic precursor.

Response: We appreciate the reviewer for the valuable comments on our manuscript. All the suggestions and comments from the reviewer have been carefully addressed and modifications have been made accordingly.

The assignment of the EXAFS spectra (Figs 3b and e) should include the fits of different scattering paths and not only the comparison with reference samples. Moreover, after analyzing the XAS results, the authors propose a dimeric NiFe structure (Figure 3f). Given that this structure is key for the following DFT calculations and reaction mechanism analysis, it would be highly enlightening if the authors include more information as to how they reached to that coordination configuration.

Response: We thank the reviewer for giving the constructive comments.

(1) As shown in Fig. R11, we optimized the fitting parameters to more accurately

visualize the coordination environment of the NiFe-BTC//G. The fits of different scattering paths and corresponding analysis are given. In addition, the EXAFS spectra of both Ni and Fe exclude significant M-M (M=Fe/Ni) metal bonds due to nanoparticles or oxides.

Fig. 11. XANES, EXAFS and FT-EXAFS spectra. (a) Ni K-edge XANES and (b) extended XANES oscillation functions $k^3 \chi(k)$ with reference samples of Ni foil, NiO. (c) EXAFS fitting curves in R space. Inset: related schematic model of Ni coordination environment in NiFe-BTC//G. (d) Fe K-edge XANES and (e) extended XANES oscillation functions $k^3 \chi(k)$ with reference samples of Fe foil, Fe_2O_3 . (f) EXAFS fitting curves in R space. Inset: related schematic model of Fe coordination environment in NiFe-BTC//G.

The above optimized fitting parameters have been included in the Fig. 3 in the revised manuscript.

(2)

Fig. R12 | Comparison between the XANES experimental spectrum and the theoretical spectrum calculated with the structure (NiFe-BTC). The K-edge theoretical XANES simulations are carried out with the FDMNES code in the framework of real-space full multiple-scattering (FMS) scheme. The spectra are convoluted by a Lorentzian function with an energy-dependent width to account for the

broadening due to the core-hole and the final state width.

In order to further elucidate the local structure of Ni and Fe, the XANES simulations, which have high sensitivity to the three-dimensional arrangement of atoms around the photoabsorber, are carried out at Ni and Fe K-edges. As shown in Figs. R12, the simulated XANES spectra based on the structure model (NiFe-BTC//G) agree well with the experimental ones. The features for the experimental spectrum are correctly reproduced. In addition, the proposed dimeric NiFe structure was proved to display a lower limiting potential according to the DFT calculations (Fig. 4 in the revised manuscript). In conclusion, according to DFT-supported guess and XAFS analysis (EXAFS fitting and XANES simulations), we suggest that the NiO₆-FeO₅ distorted octahedral species are the most possible atomic structure for NiFe species in NiFe-BTC//G.

The above XANES simulations have been included in the Supplementary Fig. 29 in the revised Supplementary Information.

As to the DFT calculations, it would be good practice to include in the supplementary document input files and optimized geometries employed. The authors should also explain their choice of cut-off energy and Monkhorst-Pack grid, either with convergence test calculations or by citing adequate literature.

Response: We are thankful to the reviewer for the valuable suggestion. The computational details, including key input parameters, have been performed in the section of Methods in our manuscripts. As suggested by the reviewer, more details were added in the revised manuscript (Page 14). In addition, more optimized structures were listed in Fig. R12.

Fig. R12 | Optimized structures of adsorbed O*, OH*, and OOH* on Ni/Fe sites.

The Monkhorst-Pack k-points were determined based on the size of unit cell. A common choice of cut-off energy and force convergence were applied following the VASP manual. In addition, we conducted further calculations with higher accuracy with higher cut-off energy and small force threshold of convergency as a test for the input parameters. As a result, less effect (< 0.02 eV on the reaction free energy) was found,

and the reliability of original choice was confirmed. More results and discussion were added in the revised manuscript (Page 14-15) and Table R2.

Table R2. Accuracy tests (reaction free energy at 1.23 V vs RHE) of cut-off energy and force convergence on reaction free energy of NiFe-BTC//G_{Ni}.

	Original	Cut-off energy test	Force convergence test		
Cut-off energy (eV)	400	450	400		
Force convergence (eV·Å ⁻¹)	0.05	0.05	0.01		
	Reaction energy (eV)	Reaction energy (eV)	Error (eV)	Reaction energy (eV)	Error (eV)
H ₂ O → (H ⁺ + e ⁻) + OH*	0.05	0.06	0.01	0.04	-0.01
OH* → (H ⁺ + e ⁻) + O*	0.17*	0.19	0.02	0.18	0.01
O* + H ₂ O → (H ⁺ + e ⁻) + OOH*	-0.28	-0.29	-0.01	-0.28	0.00
OOH* → (H ⁺ + e ⁻) + O ₂ + *	0.06	0.04	-0.02	0.06	0.00

* The bold numbers refer to the limiting-energies, namely DFT calculated limiting potential.

The above Fig. R12 and Table R2 have also been included in the Supplementary Fig. 44 and Supplementary Table 10 in the revised Supplementary Information.

The study focuses on the MOF formed by BTC but also presents results with different MOF structures that show worse performance (Supplementary Fig. 13). It would be important that the authors give us some explanation about the higher overpotentials observed with MOFs formed with other organic precursors than BTC. Did the authors perform DFT calculations with the other MOFs to account for the experimental results obtained?

Response: We thank the reviewer for raising the issue. Further DFT calculations were performed to study the effect on OER activities from different MOFs structures, including NDC, BDC, and CTC. The free energy diagrams were constructed for OER on confined NiFe-NDC, NiFe-BDC, and NiFe-CTC (Fig. R13a-c), where the Ni site was considered as the active site based on the result from NiFe-BTC in our work. An excellent agreement between DFT calculated limiting potential and experimental overpotential (Supplementary Fig. 19) was found (Fig. R13d), which indicates again the outstanding OER activity on confined NiFe-BTC. More discussions were added in the revised manuscript (Page 5) and included in Supplementary Fig. 40 in the revised

Supplementary Information.

Fig. R13 | Free energy diagram of OER on (a) NiFe-NDC//G_{Ni}, (b) NiFe-BDC//G_{Ni}, and (c) NiFe-CTC//G_{Ni}. (d) The comparison between experimental overpotential and DFT calculated limiting potential for OER with different MOF structures.

The experimental section explains the synthesis procedure of the intercalated NiFe-MOF//G. It would be important to know what the authors refer to with “The as-obtained graphite foil was immersed into DI-water and slightly sonicated for 30 min”.

Response: In our synthesis, the dilated graphite foil was immersed into DI-water and sonicated for 30 min at an ultrasonic frequency of 20 KHz, which could help to remove the excessive organic 1,3,5-benzenetricarboxylic trisodium salt on the surface of graphite foil as well as small pieces of deciduous graphite flakes.

Considering the reviewer’s concern, we modified the related text in the revised manuscript (Page 13) “The as-obtained graphite foil was immersed into DI-water and slightly sonicated at an ultrasonic frequency of 20 KHz for 30 min to remove the excessive organic 1,3,5-benzenetricarboxylic trisodium salt on the surface of graphite foil as well as small pieces of deciduous graphite flakes.”

Reviewer #4 (Remarks to the Author):

The authors present a detailed study of a hybrid electrocatalytic material built from graphite with intercalated organic complexes of Ni/Fe. The manuscript describes the synthesis, structural characterization via XPS, characterization of electro catalytic

activity via cyclic voltammeter and chronopotentiometric testing, and computational characterization via DFT simulations. The material is found to have a significantly low over potential for the oxygen evolution reaction (OER). Overall the results seem interesting and of potential high impact. However, a few aspects of the description and of the analysis of the results appear unclear or incomplete.

Response: We appreciate the reviewer for the valuable comments. We have carefully modified the manuscript and Supplementary Information according to your constructive comments.

* The authors refer to the material as confined MOFs and this is what Figure 1, in particular 1b would suggest. However, the later discussion identifies the material as small complexes of organic ligands and metal ions, embedded within the graphite layers. I would expect MOFs to be associated with crystalline porous materials, rather than molecular-level complexes.

Response: We are thankful to the reviewer for the thoughtful comment. In this work, several MOFs (including both crystalline and amorphous) of different structures are successfully confined in graphite layers and thus OER performances are also enhanced significantly (Supplementary Fig. 19). Although, the NiFe-BTC//G proposed in our work is obtained as amorphous MOFs due to the random coordination of three carboxy groups with Ni or Fe metal ions. In references, we note that amorphous MOFs are also used frequently as electrocatalysts (Chem. Rev., 2022, 122: 4163-4203; Nat Commun., 2021, 12: 4097; Angew. Chem. Int. Ed., 2020, 59: 3630-3637; Angew. Chem. Int. Ed., 2021, 60: 6362-6366).

* Connected to the use of the MOF terminology the authors correctly point out that some of the limitations of MOFs is the poor conductivity and poor mass transport during catalysis. They also identify porosity and high surface area as one of the advantages. However, in the proposed material the structural porosity of MOFs is clearly not an issue. Mass transport within graphite sheets is not explicitly discussed, although the experimental results would suggest it is not a limiting factor for this material. If I understand correctly the strategy proposed by the authors is to select promising combinations of organic ligands and metal centers from existing MOFS and intercalate them within graphite.

Response: We are thankful to the reviewer for the thoughtful comment. We totally agree that porosity and surface area are important evaluation factors for traditional MOFs which could influence the mass transport. In our work, the MOF layers are intercalated into graphite layers, which leads to a significant decrease of mass transport distance and thus the increase of OER activity. Due to the nanoconfinement from graphene multilayers, the electrocatalytic performances of all tested MOFs are

increased by our strategy (Supplementary Fig. 13), and the influences of porosity and surface area are weakened.

Regarding the reviewer's concern, we added the following discussion in our revised manuscript in Page 5 "The confinement environment endows the NiFe-MOF/G-2h electrode stabilized active sites, enhanced electrical conductivity and greatly reduced mass transport lengths. Laying the Fick's law side-by-side with the more comprehensive reports, one can indicate that the concentration of intermediates become gradient in the confinement environments during OER, which propels the intermediates from the center to the boundaries with abundant active sites (Nat Catal., 2018, 1: 135-140; Nat Commun., 2021, 12: 4294; Angew. Chem. Int. Ed., 2019, 58: 8134-8138)."

* While the experiments seem to suggest that this is a successful strategy, as the complexes in exam become more active, a rigorous explanation on why the structures are more reactive is not present. From a computational perspective, inclusion of solvent molecules in the simulations is probably very relevant. The authors adopt continuum embedding as a correction, which I assume is computed on more ideal structures of intermediates and added a posteriori to each specific simulation. This is a significant simplification on top of an already simplified approach. Moreover, an accurate description of the effect of confinement from graphite would strongly depend on the positions of the first or second solvation layer.

Response: We are thankful to the reviewer for the valuable suggestion. Indeed, we used specific solvent corrections on different intermediates through an implicit model approaching. We also agree that the solvent effect is highly depended on different kind of system (i.e. confinement environment or open system). However, the diversity of water structure makes it still a challenge to obtain an absolutely rigorous energy change from the solvent effect. Therefore, implicit models were often used as an approximation with less uncertainty. Nevertheless, we performed more comparable investigations on the solvent effect by implicit/explicit models on either open or confined systems. The explicit water structures were added directly on top of the adsorbate for the open system, while solvated water structures were used on the side of the adsorbate in the confinement environment (Fig. R14). As a result, the solvent effect on adsorption energies and reaction energies were calculated and shown in Table R3.

Fig. R14 | Explicit water model used for the solvent effect calculation with open system (a – c for O*, OH*, and OOH*) and confined system (d - f for O*, OH*, and OOH*)

Table R3. Solvent effect (eV) by implicit/explicit models on either open or confined systems.

	Implicit model on open system	Explicit model on open system	Implicit model on confined system	Explicit model on confined system
O*	0.12	0.09	0.14	0.18
OH*	0.14	0.12	0.14	0.20
OOH*	0.42	0.40	0.34	0.24

It is found that, the solvent effect on the open system using implicit model was confirmed to show its reliability by the explicit model, displaying an error less than 0.03 eV. For the solvent effect for the confined system, a similar energy change was found on O* and OH* with also a different less than 0.02 eV compared with that on open system, while larger difference was found for that on OOH*. It may be caused by the larger size of OOH*, which was more affected in the confinement environment. Moreover, an error of around 0.1 eV was obtained on the solvent effect using implicit or explicit model in the confinement effect, which was caused by the local water structures for the explicit model.

We agree that a rigorous investigation for the solvent effect is of great importance. However, the approach of solvent effect used in this work is still reliable, for the trend over different materials. We added more discussions in the revised Supplementary

Information (Supplementary Fig. 41 and Supplementary Table 11).

While some of the computational details are missing or not properly justified (revPBE may not be the most appropriate functional, how is spin magnetization handled, is the graphite layer allowed to relax, etc.), it is true that they may be less relevant in a comparative analysis. However, appropriate accounting of hydrogen bonds and statistical sampling may significantly alter the computed free energy diagrams. Moreover, it would be useful to understand if the structure of the confined solvent contributes to the enhanced reactivity and in which way it affects mass transport within the graphite layers.

Response: Thank you for the comment. We agree that the functional do affect the absolute energy value in the calculation. However, rPBE has been already used in many previous works for OER or ORR, which is still appropriate especially for relative trend (Nat. Catal. 2022, 5: 109-118; J. Phys. Chem. C 2020, 124: 25796-25804). The major reason is the scaling relation based on GGA and GGA+vdw is comparable (PCCP, 2020, 22, 5293-5300). The effect from spin magnetization was also tested, showing a small error (< 0.04 eV) on the limiting energy. The graphite layer was allowed to relax. We added more details in the Supplementary Information (Supplementary Table 12).

Table R4. Test of energy effect from spin magnetization on NiFe-BTC//G_{Ni}.

	Reaction energy (eV)	
	Spin-off	Spin-on
$\text{H}_2\text{O} \rightarrow (\text{H}^+ + \text{e}^-) + \text{OH}^*$	0.05	0.07
$\text{OH}^* \rightarrow (\text{H}^+ + \text{e}^-) + \text{O}^*$	0.17*	0.21
$\text{O}^* + \text{H}_2\text{O} \rightarrow (\text{H}^+ + \text{e}^-) + \text{OOH}^*$	-0.28	-0.14
$\text{OOH}^* \rightarrow (\text{H}^+ + \text{e}^-) + \text{O}_2 + *$	0.06	-0.14

* The bold numbers refer to the limiting-energies, namely DFT calculated limiting potential.

As discussed above, these tests were confirmed to show small effect for a comparative analysis of trend.

* The structures of the simulated intermediates should be included in the supplementary information, as they would allow to evaluate the type of catalytic pathways that have been explicitly analyzed. My understanding of the computational details suggests that the authors are using the Computational Hydrogen Electrode approach to compute the effect of the potential on the catalytic activity, references to the appropriate literature would be important. It is also not clear if the calculations reported for the non-confined MOFs are computed for bulk 3D materials or for small molecular complexes without

the graphite layers (as visualized in Figure 24)?

Response: We fully appreciate the professional and constructive comments from the reviewer. A schematic diagram with intermediates to show the reaction path was added as follows:

Fig. 15 | Schematic diagram of reaction pathway for OER on NiFe-BTC_{Ni} as an example.

Indeed, the CHE model was applied, which was also described in the section of “Reaction free energy calculations” in the Supplementary Information. The reference was added in the revised manuscript. MOF structures with single unit was applied for all the calculations. In addition, we performed further calculations to investigate the effects from 3D structures with the construction of more NiFe units linked with BTC structures (Fig. R 16). Table R5 shows the energy calculated based on more NiFe units, where a small effect was also found on the reaction free energy.

Fig. R16 | (a) Structure of NiFe-BTC//G. (b) Structure of NiFe-BTC//G with more NiFe units linked on

the other side of BTC structures.

Table R5. Test of energy effect from varying NiFe units linked with BTC on NiFe-BTC//G_{Ni}.

	Reaction energy (eV)	
	One NiFe unit	Three NiFe units
$\text{H}_2\text{O} \rightarrow (\text{H}^+ + \text{e}^-) + \text{OH}^*$	0.05	-0.01
$\text{OH}^* \rightarrow (\text{H}^+ + \text{e}^-) + \text{O}^*$	0.17*	0.19
$\text{O}^* + \text{H}_2\text{O} \rightarrow (\text{H}^+ + \text{e}^-) + \text{OOH}^*$	-0.28	-0.23
$\text{OOH}^* \rightarrow (\text{H}^+ + \text{e}^-) + \text{O}_2 + *$	0.06	0.05

* The bold numbers refer to the limiting-energies, namely DFT calculated limiting potential.

More discussions were added in the revised manuscript and included in Supplementary Figs. 42 and 43, as well as Supplementary Table 13 in the revised Supplementary Information.

As a very minor note, in the Computational Details of the main manuscript the authors refer twice to supplementary Figure 23 instead of 24.

Provided the points above are fully addressed, I would recommend publication of the manuscript on Nature Communications.

Response: We are thankful to the reviewer for the kind recommendation and careful review. The false supplementary Figure in the main manuscript was amended. The revised manuscript and Supplementary Information were both carefully checked to avoid similar errors.

REVIEWERS' COMMENTS

Reviewer #1 (Remarks to the Author):

Authors addressed my previous concern and the manuscript is ready to be published.

Reviewer #2 (Remarks to the Author):

The authors made significant efforts in responding to the comments and in modifying the manuscript. I, therefore, recommend publishing the manuscript in Nature Communications.

Reviewer #3 (Remarks to the Author):

Given that the arised issues were fully addressed and the article and the SI were consequently modified and improved, I would recommend this article to be published on Nature Communications.

Reviewer #4 (Remarks to the Author):

The authors addressed in a positive and clear way most of the comments and remarks of the reviewers. The amount of details in the supporting information is now adequate. I recommend publication on Nature Communications.